# Multi-Channel Electrical Discharge Machining of Ti-6Al-4V Enabled by Semiconductor Potential Differences

**DOI:** 10.3390/mi16020147

**Published:** 2025-01-26

**Authors:** Xuyang Zhu, Tao Wei, Sipei Li, Guangxian Li, Songlin Ding

**Affiliations:** 1School of Engineering, RMIT University, Melbourne, VIC 3083, Australia; s3615172@student.rmit.edu.au (X.Z.); s3939148@student.rmit.edu.au (T.W.); 2Faculty of Education, Monash University, Melbourne, VIC 3800, Australia; slii0259@student.monash.edu; 3School of Mechanical Engineering, Guangxi University, Nanning 530004, China

**Keywords:** silicon, electrodes, electric discharge machining, EDM, multi-channel, potential differentials

## Abstract

Titanium alloys are difficult to machine using conventional metal cutting methods due to their low thermal conductivity and high chemical reactivity. This study explores the new multi-channel discharge machining of Ti-6Al-4V using silicon electrodes, leveraging their internal resistivity to generate potential differences for multi-channel discharges. To investigate the underlying machining mechanism, the equivalent circuit model was developed and a theoretical simulation was carried out. Comparative experiments with silicon and conventional copper electrodes under identical parameters were also conducted to analyze discharge waveforms, material removal rate, surface quality, and heat-affected zones (HAZ). The results demonstrate that the bulk resistance of silicon is the main mechanism for generating multi-channel discharges. This process efficiently disperses the discharge energy of the single discharge pulse, resulting in smaller craters, smoother machined surfaces, and shallower recast layers and HAZ.

## 1. Introduction

Electrical discharge machining (EDM) is a nonconventional method that has been extensively used in industry for the machining of difficult-to-cut materials, such as titanium alloys [1], nickel-based alloys [2], composite materials [3], and polycrystalline diamond (PCD) [4,5]. As a thermal machining process, EDM utilizes high-temperature plasma generated by pulsed voltage between the electrode and the workpiece to remove workpiece materials. Within the plasma channel, the temperature is in the range of approximately 8000–12,000 K. This high temperature can result in defects on the workpiece surface, for example, rough surface finish, micro-cracks, thick recast layer, and high residual stress, which significantly affect the mechanical properties of the machined surface, such as wear resistance and fatigue strength [6,7]. Nevertheless, in some specific applications, a rough surface is desirable in order to improve the hydrophobic properties of the surface, i.e., to change the wettability and lubricity of the surface [8].

Compared to conventional CNC machining, or nonconventional methods such as laser machining and other laser-assisted manufacturing methods [9], the efficiency of EDM is low due to the low overall energy provided by the interrupted or discontinuous discharge pulses [10]. To increase the machining efficiency, larger discharge energy has to be adopted through high discharge current and longer time on. However, the increase in discharge energy can induce even severe thermal defects, eventually deteriorating the quality of the workpiece surface. Therefore, optimizing the distribution of discharge energy during the machining process to achieve finer surface finishes has become an urgent challenge to address.

Multi-channel discharging EDM is a new concept aiming to address the issue by distributing the energy of one single spark through several channels. Typically, each pulse generated by the power supply generates only one spark in conventional EDM, and the quality of the machine surface is influenced by the energy of the spark, which can be controlled by discharge current in prior [11], voltage, polarity, duty cycle (time on/off) [12], and properties of dielectric fluid [13]. If the energy in a single pulse can be distributed into multiple discharge channels at different discharging positions simultaneously, and the reduced peak current in each channel can limit the thermal defects, the machine efficiency can be maintained because the total discharge energy has not been reduced. Therefore, multi-channel discharge offers a viable approach to addressing the challenge of achieving high surface quality under conditions of high discharge energy.

Currently, the generation of multi-channel discharges in EDM can be categorized into three methods. The most widely used method is the application of multiple electrodes. Kunieda and Muto [14] utilized a two-electrode series with a polar adaptive pulse generator to equalize the gap voltage, generating two-channel discharges. The material removal rate (MRR) was increased, and both surface roughness (SR) and electrode wear were reduced. Yilmaz and Okka [15] utilized multiple electrodes to machine Inconel 718 and Ti-6Al-4V. Their results show that the single-electrode method offered relatively higher MRRs and lower electrode wear rates, while the multiple-electrode method produced superior surface roughness. Similar experiments were also conducted by Bozdana and Ulutas [16]. The drilling of Inconel 718 via multiple electrodes reduced machining time, and dimensional accuracy and surface quality were significantly improved. Yu et al. [17] employed discrete electrodes for multi-channel discharge machining. The multi-point ablation enlarged ablation range and dispersed discharge energy. Therefore, both MRR and surface finish of the proposed multi-channel EDM method were improved. However, a large number of resistors were utilized to achieve multi-channel discharges in their experiments, which compromised the power supply’s efficiency. Deng et al. [18] optimized the distribution of inter-electrode voltage by modifying the discharge circuit, thereby increasing the occurrence probability of multi-spark discharge. A 15.8% reduction in surface roughness was achieved across several ultra-thick cutting workpieces. Fu et al. [19] adopted a similar method to machine Inconel 718, and there was an 85.2% increase in machining efficiency and a 16.6% reduction in surface roughness. Also, micro-cracks and thickness of recast layer were reduced. Yang et al. [20] developed a multi-channel EDM approach using a capacitive coupling method: by connecting isolation capacitors in series with the workpiece, multiple parallel discharge gaps were established, enabling the formation of multi-channel discharges. In addition, a compensating capacitance was connected in parallel with the pulse generator to shorten the balance restoring time of discharge circuits. The results demonstrated that MRR was improved by increasing compensating capacitance and number of electrodes. However, the increase in compensation capacitance meant altering the structure of pulsed power supply, which caused the practical applicability of this method to be restricted by factors such as specifications of electrodes and output channels of the power supply.

In addition to the “multi-electrode” method, some studies attempt the powder-mix electrical discharge machining (PMEDM) to generate multi-channel discharges by adding conductive powders in the dielectric. This technique utilizes powder additives to alter the breakdown strength between the electrode and the workpiece, reduce the discharge gap, and create a bridging effect [21]. Initially, studies involving the addition of suspended particles into the dielectric were aimed at improving SR, MRR, and tool wear rate (TWR) during the machining of titanium alloys. Ming and He [22], in their PMEDM experiments, observed that the addition of powder caused the discharge energy of primary sparks to be distributed, compensating for the occurrence of secondary discharges with lower energy. This reduced the longitudinal depth of craters while increasing the formation of shallow pits in the transverse direction. Similarly, Shabgard and Khosrozadeh [23] presented comparable findings by introducing carbon nanotube (CNT) into the dielectric fluid. Their results show that individual sparks could discharge with low energy intensity at multiple locations. However, neither study provided conclusive evidence for the existence of multi-channel discharges; they only identified secondary discharges and a reduction in single discharge energy intensity. Building on this, Wang et al. [24] achieved a multi-channel discharge process by mixing silicon powder into the dielectric, and they pointed out that the none equipotential characteristic of semiconductor particles caused multi-channel discharges and dispersion of discharge energy.

Multi-channel discharges can be generated via the application of a semiconductor as an electrode [25,26]. Chen et al. [27] investigated the discharging characteristics of semiconductors in wire-cut EDM and suggested that the secondary discharge was triggered by potential differences after the first discharge due to body resistance and a contact potential barrier. Zhang et al. [28] discovered that multi-channel discharges occurred in the wire-cut EDM with an ultra-fine molybdenum wire electrode (0.05 mm, 28.5 Ω·m). It was found that multi-channel discharges effectively distributed the discharge energy, resulting in a reduction in surface roughness (*R_a_*) of 37.7–46.6% and an improvement in machining efficiency of 8.7–15.6%. Other multi-channel discharge approaches were also proposed in some research. For example, some studies claimed that multi-channel discharge may occur by reducing the gap distance, but this phenomenon has not been repeated or verified by any other studies [29].

Understanding the mechanisms of various multi-channel EDM processes is critical for the industrial application of multi-channel discharge machining of ultra-hard alloys. However, existing research has primarily focused on multi-electrode and PMEDM, with limited studies addressing multi-channel discharges based on potential differences. This study investigates the operating mechanism of single-electrode multi-channel discharges, leveraging the unique electrical properties of semiconductors. The findings reveal that the potential difference created by the bulk resistivity of silicon electrodes provides an effective means to achieve multi-channel discharges. Post-machining analyses of discharge waveforms, surface quality, elemental composition, and HAZ depth were conducted to compare the practical performance of multi-channel and single-channel discharge machining.

## 2. Materials and Methods

The sample workpieces are Grade 5 Ti-6Al-4V bars (Specialty Metals Ltd., Welshpool, Australia, Table 1), with a diameter of *Φ* 9.53 mm and 25 cm in length. The electrode is a rectangular single-crystal silicon sheet with dimensions of 15 mm (width) × 60 mm (length), resistivity 1 Ω·cm. In addition, a copper electrode of the same size was used as comparison. Due to the transient nature of discharge and the randomness of discharging locations, it is difficult to accurately capture the potential at the discharge point on the electrode surface during the EDM process. To overcome this limitation, two independent discharge channels were devised to examine the number of simultaneously discharged channels, as illustrated in Figure 1. The experimental setup encompassed an oscilloscope (Gwinstek GDS-2104, Good Will Instrument Co., Ltd., Taiwan, ROC), a voltage probe (Rigol RP2200, Rigol Technologies, Beijing, China), a current probe, an amplifier (Tektronix TCP303/TCPA300, Tektronix, Beaverton, OR, USA), and a die/sink EDM machine (CHMER 50MP Ching Hung Machinery & Electric Industrial Co., Ltd., Taiwan, ROC). Once a plasma channel is formed between the electrode and the workpiece surface, the discharge current and the voltage signal are captured by the current probe and the voltage probe, respectively. The signals are then acquired and displayed by the oscilloscope. The status of the discharge channel could be directly assessed by evaluating the currents in both current channels within the same voltage pulse. The schematic diagram of the system is shown in Figure 2.

The machining parameters used in the experiments are listed in Table 2. These parameters were selected based on preliminary experiments to ensure that the chosen settings prevent electrode damage caused by excessive current while maximizing the manifestation of multi-channel discharge phenomena induced by the semiconductor bulk resistance. Since the response time of the current sensor, amplifier, and oscilloscope is at a scale of ns, which is significantly smaller than the pulse width, the two signals can be collected simultaneously.

## 3. Experiment Results

### 3.1. Material Removal Rate (MRR) and Electrode Wear Rate (EWR)

The MRR and EWR are determined by the loss of weight after the EDM, which can be calculated via following equations:(1)$$MRR=\frac{\left(W_{b}-W_{a}\right)*1000}{T}$$(2)$$EWR=\frac{\left(T_{b}-T_{a}\right)*1000}{T}$$
where *W_b_* and *W_a_* are weights (g) of the workpiece before and after EDM, *T_b_* and *T_a_* are weights (g) of the electrode before and after EDM, and *T* is the duration of EDM (min).

Figure 3 shows the comparison of MRRs and EWRs when using silicon and copper as the cathode, respectively. MRR using silicon electrodes is smaller than that of using copper electrodes due to the consumption of electrical energy because of the inherent body resistance of silicon. However, the difference between MRRs of the two workpieces is obvious when using Cu electrodes. This is because copper electrodes cannot generate multi-channel discharges, sparks are generated between electrode pairs of Cu/1st workpiece and Cu/2nd workpiece in turn. In contrast, when using silicon electrodes, the capability for multi-channel discharge enables simultaneous machining on both workpieces. As a result, no significant difference in MRR was observed between the two workpieces. With regard to EWR, the wear rate of silicon electrode is five times higher than that of copper electrode. This discrepancy is partially attributed to the brittle nature of the silicon electrode where higher material removal rates were observed due to material fracture during the machining process, similar to those found in [30,31].

### 3.2. Discharge Waveform

Due to different discharging mechanisms, waveforms in EDM processes using silicon and copper electrodes are different. Specifically, the discharges in silicon-electrode EDM present multi-channel waveforms (Figure 4a). Discharge waveforms simultaneously appear in the two current channels CH2 and CH3, indicating the occurrence of dual-channel discharge. For each pulse (Figure 4b), the current waveform exhibits a distinctive shape of smooth tapering. The current of CH2 gradually decreases, whereas the current of CH3 increases slightly, which indicates the constant total energy of one discharge. In the copper-anode EDM, the waveforms present typical characteristics of single discharge (Figure 5): the current changes in only one channel (CH2). Furthermore, the pulse voltage of copper-anode EDM consists of higher open-circuit voltage with longer duration and lower breakdown voltage, and there is a delay for the surging of current, which indicates the breakdown of dielectric. In contrast, the voltage of Si-anode EDM does not show a drastic decrease due to the breakdown of the dielectric, which is almost constant throughout the discharging, and the current rising simultaneously with the voltage pulse.

### 3.3. Workpiece Surface Morphology

A comparison of the morphology of machined surfaces is shown in Figure 6. It can be observed that the surface roughness *R_a_* of the copper electrode (*R_a_*: 1.7228 μm) is significantly higher than that of the silicon electrode (*R_a_*: 0.89405 μm). According to the model in the study of Qin et al. [32], the diameter of plasma (*D*) increases with the increase in discharge current (*I*), as shown in the following equation:(3)$$D=102.5\times 10^{-6}\cdot I^{0.4746}$$

According to current wave forms, the generation of multi-channel discharges using the silicon electrode dispatch energy to different channels, which reduces the current per channel. Correspondingly, the reduced current leads to smaller pits and better surface quality (*R_a_*: 0.89405 μm) when using silicon anode, which is half of the *R_a_* achieved with copper electrode.

Figure 7 and Figure 8 show the microstructure of the Ti-6Al-4V workpiece after EDM with anodes of silicon and copper, respectively. Specifically, lower current in multi-channel discharging in the Si-anode EDM reduced the explosive force, contributing to less reconsolidated materials on the machined surface. In comparison, the machined surface using the copper anode is covered by a larger area of resolidified material, with more pronounced impact phenomena. The edges of individual craters are sharp and exhibit distinct protrusions. In contrast, the craters formed with the silicon electrode have smoother edges, smaller individual crater areas, and lack prominent protrusive features. Energy dispersive spectroscopy (EDS) was used to analyze the chemical composition on machined surfaces. In both experiments, electrode material migration was observed. From the EDS analysis results shown in Figure 9 and Table 3, it can be concluded that, regardless of whether copper or silicon electrodes were used, the workpiece surface exhibited attachment of migrated electrode material. However, the proportion of silicon migration was approximately twice that of copper, which aligns with the significant consumption of silicon electrodes during the process.

Additionally, there was a notable difference in the vanadium (V) content on the workpiece surface between the two electrodes. Specifically, the V content after processing with silicon electrodes was only half of that observed with copper electrodes. This could be attributed to the high vapor pressure of vanadium at elevated temperatures, which makes it more prone to volatilization or migration during the discharge process. The high temperatures and possible chemical reactions associated with silicon electrodes may have further accelerated this phenomenon, resulting in a significant reduction in V content.

For aluminum and titanium, no significant differences in surface content were observed between the two types of electrode.

The migration of electrode material during the machining process modifies the workpiece surface condition, resulting in altered chemical and physical properties that significantly impact the material’s performance compared to its pre-machined state. For example, after machining with copper electrodes, the thermal and electrical conductivity of Ti-6Al-4V improves, and copper’s excellent antibacterial properties make it advantageous for applications in medical and implantable devices [33], although its susceptibility to electrochemical corrosion in acidic or saline environments limits its use in marine or chemical processing equipment. On the other hand, machining with silicon electrodes enhances the hardness and wear resistance of the material [34], favoring its application in cutting tools for aerospace components [35]. However, the incorporation of silicon increases surface brittleness, restricting its suitability for applications requiring high toughness and impact resistance.

Worn areas on surfaces of Si and Cu electrodes are examined via SEM, and worn morphology of the Si anode (Figure 10) differ significantly from that of the Cu anode (Figure 11). The worn Si anode exhibits a rough texture with distinct fracture edges resembling a coral-like structure, along with small pores and collapse marks. Considering the brittle nature of silicon and the high electrode wear rate, it is reasonable to infer that the wear primarily results from electrode fractures during machining. In contrast, the worn area of the copper electrode is relatively smooth with small voids and localized material re-deposition. This stark difference highlights the distinct behaviors of silicon and copper as anodes in EDM processes.

Titanium alloys, due to their low thermal conductivity, exhibit distinct subsurface characteristics after EDM compared to other alloys [36,37,38]. Figure 12a shows the cross-sectional view of Ti-6Al-4V processed using a copper electrode. After machining, the total affected depth measures approximately 21.1 μm. After etching, the thickness of the HAZ and the recast layer is clearly distinguishable, revealing three regions: (I) the recast layer, (II) the HAZ between the recast layer and the substrate material, and (III) the unaffected base material.

Figure 12b illustrates the cross-sectional view of Ti-6Al-4V processed using a silicon electrode. The total affected depth is approximately 16.1 μm, with well-defined boundaries between the regions. The recast layer shows visible dendritic structures at deeper positions, while the layer processed with a copper electrode primarily exhibits coarse equiaxed grains and fine surface grains, similar to those found in [39].

## 4. Discussion

In conventional EDM systems, high-conductivity materials like graphite and copper are typically used as electrodes due to their minimal internal bulk resistance and lack of additional impedance in the discharge circuit, allowing the full release of pulse energy through the plasma discharge channel in a single pulse. In contrast, using semiconductors as electrodes introduces bulk resistance, contact barriers, and non-isotropic characteristics, which hinder the complete release of energy through a single channel. This setup enables the phenomenon of “multi-channel discharging”, where a second breakdown can occur at a location away from the initial discharge spot due to a local high inter-electrode voltage. Multi-channel discharge disperses the energy released in a single discharge among multiple discharges, though not always evenly, with impedance playing a key role in its occurrence.

### 4.1. Equivalent Circuit Model of Single-Electrode Multi-Channel Discharge

Figure 13 illustrates the instantaneous state of three discharge channels at a specific moment under a single voltage pulse when using a silicon electrode. Since the three discharge channels do not break down simultaneously, their discharge states differ at any given moment. Assuming the discharge channel states at a specific moment are presented, with the breakdown sequence occurring sequentially from left to right, the states of these different discharge channels at this moment can be described as follows: (I) the discharge channel that has completed its discharge begins to expand; (II) ions and electrons within the discharge channel are actively moving; and (III) the plasma that has just undergone breakdown is in the early stages of formation.

Previous studies on multi-channel discharge typically involved the use of multi-electrode configurations. In such systems, the equivalent circuit assumes a fixed resistance for each discharge channel. As shown in Figure 14a. The impedance of each discharge channel is *R*_0_ + *R_n_*, where *R_n_* is the electrode resistance, and the potential difference between electrodes is zero when all electrodes share the same resistance. In contrast, in a single-electrode discharge circuit, the resistance of each discharge channel varies. As illustrated in Figure 14b, the electrode is divided along the *X*-axis, with the resistance of discharge channels *P*_1_, *P*_2_, and *P_n_* being *R*_1,_
*R*_1+2_, and *R*_1+2+…+*n*_, respectively. The potential difference between discharge positions arises from the resistance difference, increasing with the distance from the discharge point to the electrode’s power connection point. Unlike conductor electrodes which fully release energy in the first discharge, semiconductor electrodes, despite their energy dissipation due to bulk resistance, can generate multiple discharge channels.

### 4.2. The Principle and Mechanism of Single-Electrode Multi-Channel Formation

The mechanism of multi-channel discharging using semiconductor electrodes can be attributed to the unique electrical properties of semiconductors, as depicted in Figure 15. Generally, the high bulk resistance of semiconductors only causes the rapid drop of gap voltage near the position of discharging, and the high electrical potential at other positions can still cause the breakdown of the dielectric, forming multiple plasma channels within a single pulse. Taking the two-channel discharging as an example, the entire discharging process consists of the following stages:

Stage 0 and 1: With a reduction in the anode/cathode gap, the strength of the electrical field increases. Once the electric field strength exceeds the dielectric strength of the fluid, a discharge channel is formed by the breakdown of dielectric and begins expanding. At this stage, significant electron and ion movements occur within the channel.

Stage 2: Due to the high resistance of the semiconductor body, the gap voltage drops slightly, and the energy of the pulse is not instantaneously released. After a short lag time, the breakdown of the dielectric happens at another location, forming the secondary discharge. The two discharge channels are co-existing and independent. As the two channels are connected in parallel, the equivalent resistance of the total circuit decreases. As the secondary discharge is established, the current of the first discharge decreases with the increase in the secondary discharge, and the gap voltage decreases as well.

Stage 3: The waveform of the current of the secondary discharge is like that of the prior discharge, whereas the duration is shorter. Although equivalent resistance of the whole circuit decreases, pulse energy and gap voltage tend to fall steadily and the whole discharge system shows a self-sustaining discharge cycle. With the end of the pulse voltage, the self-sustained discharge state is broken. The positively and negatively charged particles in the discharge channel combine into neutral particles and the breakdown of dielectric finishes.

### 4.3. Voltage Difference Between Electrodes

To further elucidate the discharge mechanism of single-electrode multi-channel EDM, a steady-state numerical simulation model of electrode voltage distribution was established. This model compares the voltage distribution within copper and silicon electrodes before and after discharge. The numerical simulation was conducted using COMSOL Multiphysics software (version 5.3), employing the DC current and electric circuit interfaces. The governing equations for the DC current interface include the relationship between electric field and potential (Equation (4)), Ohm’s law (Equation (5)), and the current continuity equation (Equation (6)):(4)E=−∇φ(5)J=σE(6)∇⋅J=0

In the equation, *J* represents current density, *E* is the electric field, *σ* is the material conductivity, and *φ* is the electric potential.

The geometrical models before and after discharge are shown in Figure 16 and Figure 17, respectively. The electrodes used in the simulation are a metallic copper electrode and a semiconductor silicon electrode, while the workpiece is a 2 mm diameter Ti-6Al-4V block. The electrode dimensions are 100 mm × 20 mm × 2 mm, connected to the power supply through conductive metal sheets. In Figure 17, the red circular area at the electrode’s bottom represents the discharge crater.

The mesh sizes before and after discharge are shown in Figure 18 and Figure 19, respectively. A swept mesh was applied before discharge, while after discharge, a swept mesh was used for the workpiece, with other domains employing free tetrahedral meshing. The mesh near the discharge channel was refined for higher accuracy.

Three reasonable assumptions were made for this model:The resistivity of the semiconductor silicon is constant.Discharge energy is evenly distributed across all discharge channels.The contact barrier between the semiconductor and metal is ignored.

Based on these assumptions, the circuit diagrams for the simulations before and after discharge are presented in Figure 20. In these diagrams, *U*_0_ represents the open-circuit voltage, and *R*_0_ denotes the current-limiting resistor. The anode’s input surface (Terminal 1), the workpiece surface (Terminals 3–6), and the discharge channel (Terminal 2) were connected to the external circuit as external terminals. The remaining external surfaces were set as electrically insulated boundary conditions. Simulation parameters are listed in Table 4.

By applying a voltage to the workpiece, the internal voltage distribution within both silicon and copper electrodes remains uniform under initial conditions (Figure 21).

However, during a single discharge event, the voltage within the copper electrode drops significantly to below 40 V (Figure 22), insufficient to sustain secondary breakdown.

In contrast, Figure 23 illustrates the internal voltage distribution of the silicon electrode during a discharge event. Unlike the copper electrode, the silicon electrode exhibits a gradual voltage drop around the discharge location due to its intrinsic bulk resistance, resulting in spherical equipotential surfaces radiating outward. The farther the location from the discharge point, the higher the voltage; the closer to the discharge channel, the steeper the voltage gradient.

If the potential difference between the silicon electrode and the workpiece at another location exceeds the breakdown voltage, a secondary discharge can occur, forming a subsequent discharge channel

## 5. Conclusions and Further Work

A novel potential-difference-based multi-channel discharge machining method was developed by using silicon semiconductor electrodes to improve the surface quality in EDM. The new theory and mechanism underlying this single-electrode multi-channel discharge method were discovered, and a significant enhancement in surface roughness was achieved. Based on the results obtained from the theoretical analysis and machining experiments, the following conclusions can be drawn:(1)Multiple discharge channels can be generated within one voltage pulse using a single semiconductor electrode, whereas only one discharge channel can be generated by the metal electrode.(2)When using semiconductor electrodes, the discharge waveforms appear at the beginning of the pulse voltage, usually in an inverted U-shape, which is different from those achieved by using metal electrodes where the current has some hysteresis and a more defined ON/OFF point.(3)In comparison to metallic copper electrodes, silicon semiconductor electrodes exhibit greater material migration and higher electrode wear.(4)The number of discharge channels in potential-difference-based multi-channel discharge is primarily determined by a combination of factors, including the open-circuit voltage, the bulk resistivity of the electrode material, contact resistance, anisotropic properties, electrode dimensions, and the electrode layout.(5)In multi-channel discharge EDM, the current is distributed over multi-channels, resulting in more dispersed discharge energy, higher surface quality of the workpiece, and smaller diameter of the corrosion pits. In single-channel EDM, the current is not distributed, which results in more concentrated discharge energy, lower surface quality of the workpiece, and larger diameter of the corrosion pits.(6)Due to the energy re-distribution in multi-channel discharge machining, the HAZ is relatively shallow, and the recast layer on the workpiece exhibits a distinct dendritic structure with a thin transition zone. In contrast, workpieces machined with single-channel discharge show a thicker HAZ, accompanied by coarse equiaxed grains and finer surface grains.

In this study, the tool wear ratio (TWR) of silicon electrodes was found to be significantly higher compared to traditional copper electrodes. Investigating the underlying mechanisms of this abnormal wear is an area that warrants further exploration. Currently, there are few studies utilizing silicon as an EDM electrode, with its primary applications still centered around the semiconductor industry. Additionally, the inherent brittleness of silicon and the challenges associated with machining it into ideal electrode shapes pose significant obstacles to its broader adoption as an electrode material. Developing methods to manufacture silicon electrodes that are both cost-effective and easily formable should be a key focus of future research.

Furthermore, when silicon migrates to the surface of Ti-6Al-4V during EDM, it forms a silicon-rich modified surface layer, thereby altering the surface composition and properties of the material. Existing studies have shown that the high temperatures generated during EDM can promote reactions between silicon and titanium, resulting in the formation of hard ceramic compounds, such as titanium silicides (e.g., TiSi_2_) [40]. Other reports suggest that this process may also affect frictional characteristics, wettability, corrosion resistance, and biocompatibility [41]. These potential benefits highlight important directions for future research as they present significant opportunities for practical applications.

## Figures and Tables

**Figure 1 micromachines-16-00147-f001:**
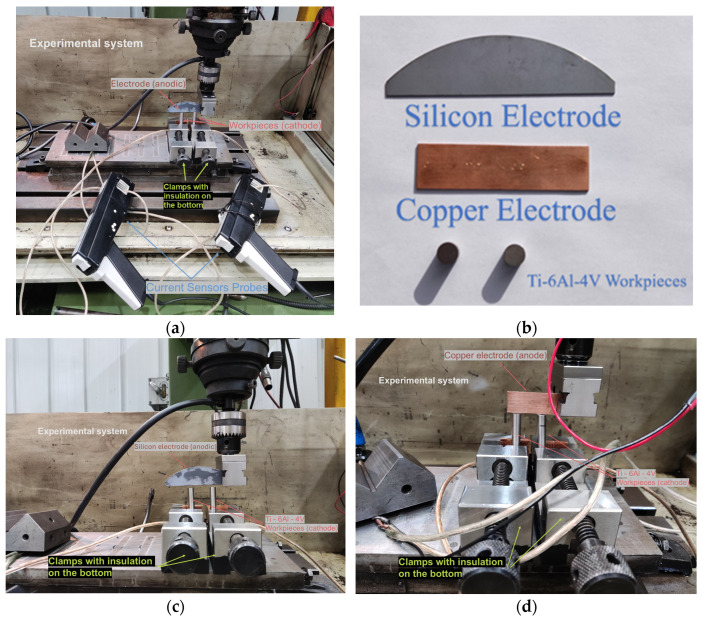
Experimental setup of electrodes and workpieces: (**a**) detection loops for current waveforms; (**b**) electrodes and workpieces used in the experiment; (**c**) multi-channel discharge experiments with a semiconductor electrode; and (**d**) single-channel discharge experiments with a copper electrode.

**Figure 2 micromachines-16-00147-f002:**
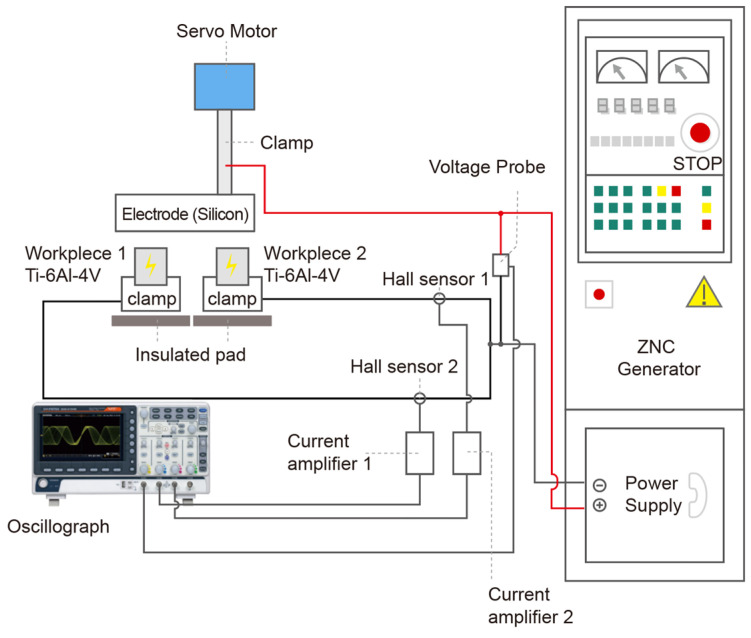
Schematic diagram of the experimental system.

**Figure 3 micromachines-16-00147-f003:**
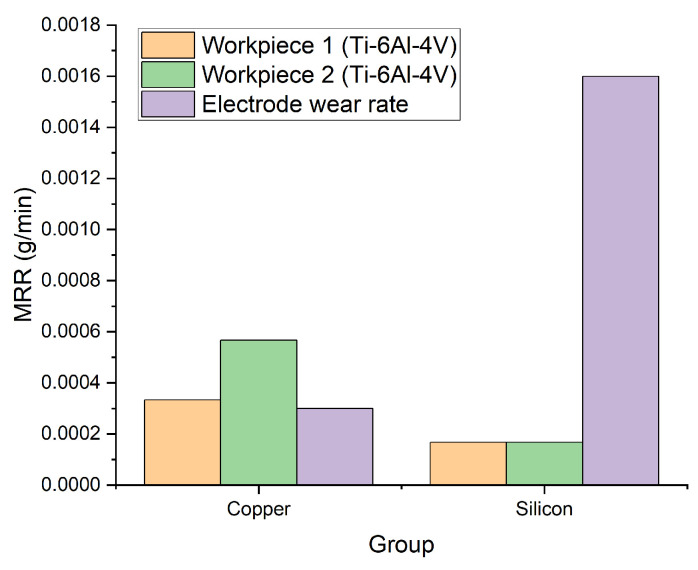
The influence of different types of electrodes on material removal rate and electrode wear rate.

**Figure 4 micromachines-16-00147-f004:**
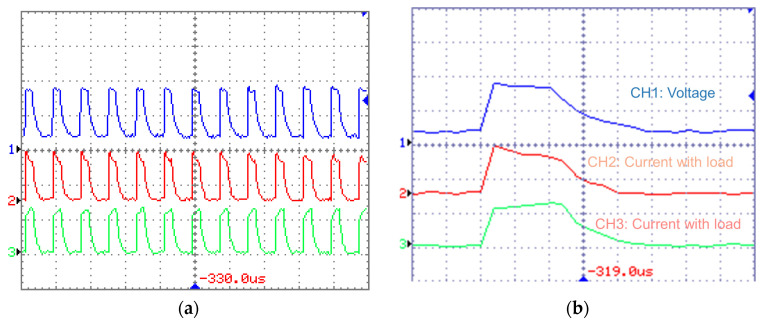
Multi-channel discharge voltage and current waveform of EDM (Electrode: Silicon) (**a**): CH1: 20 V/div CH2: 1 A/div CH3: 1 A/div M: 50 μs Trigger (T): CH1 (**b**): CH1: 20 V CH2: 1 A CH3: 1 A M: 5 μs T: CH1.

**Figure 5 micromachines-16-00147-f005:**
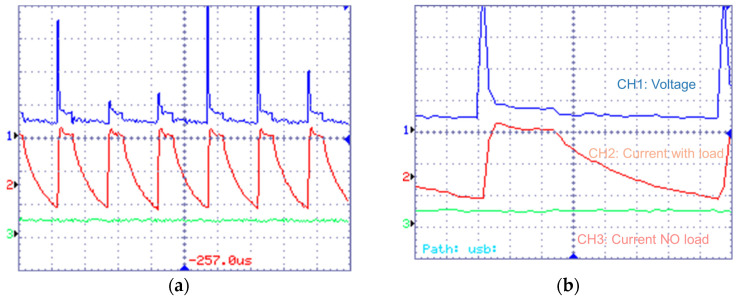
Single-channel discharge voltage and current waveform of EDM (Electrode: Copper) (**a**): CH1: 20 V/div CH2: 1 A/div CH3: 1 A/div M: 50 μs Trigger (T): CH1 (**b**): CH1: 20 V CH2: 1 A CH3: 1 A M: 5 μs T: CH1.

**Figure 6 micromachines-16-00147-f006:**
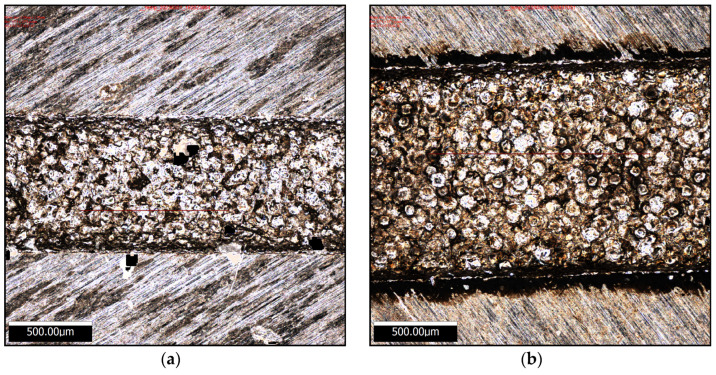
Structure of different workpiece surfaces observed using the optical microscope Alicona: (**a**) workpiece surfaces machined with silicon electrode *R_a_*: 0.89405 μm; and (**b**) workpiece surfaces machined with copper electrodes *R_a_*: 1.7228 μm.

**Figure 7 micromachines-16-00147-f007:**
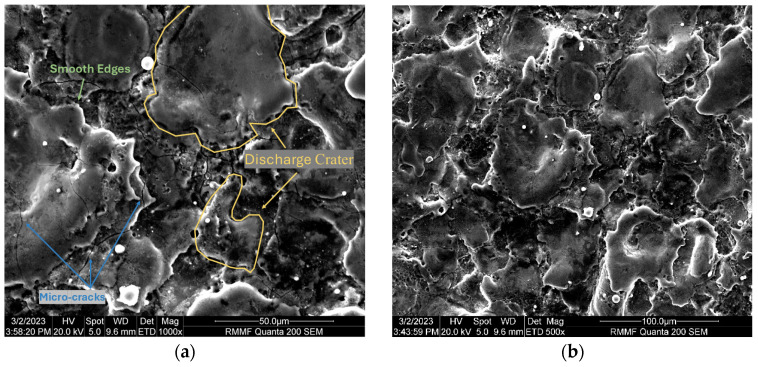
SEM micrographs of the surface of a workpiece after EDM processing using Si electrodes: (**a**) 1000× magnification; and (**b**) 500× magnification.

**Figure 8 micromachines-16-00147-f008:**
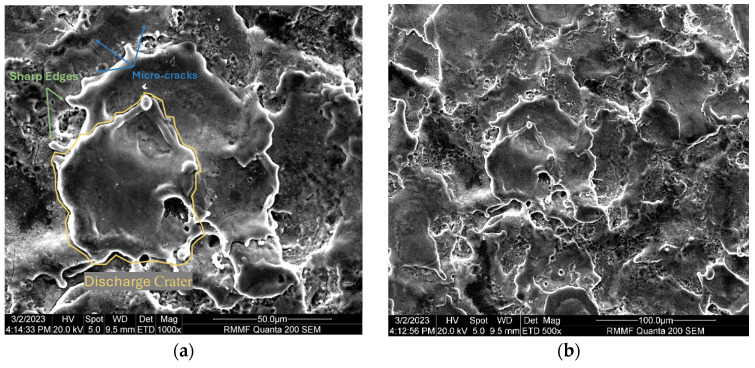
SEM micrograph of the surface of a workpiece after EDM processing using copper electrodes: (**a**) 1000× magnification; and (**b**) 500× magnification.

**Figure 9 micromachines-16-00147-f009:**
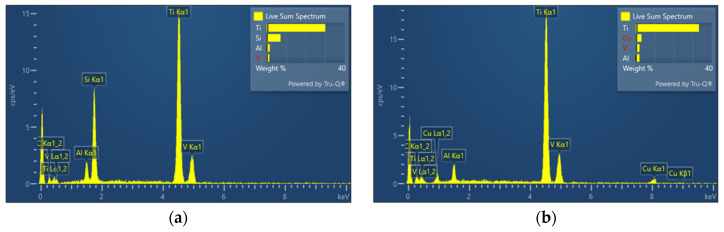
Energy dispersive spectroscopy analysis of the surface of the workpiece after EDM with different electrodes: (**a**) energy dispersive spectroscopy using silicon electrode: and (**b**) energy dispersive spectroscopy using copper electrode.

**Figure 10 micromachines-16-00147-f010:**
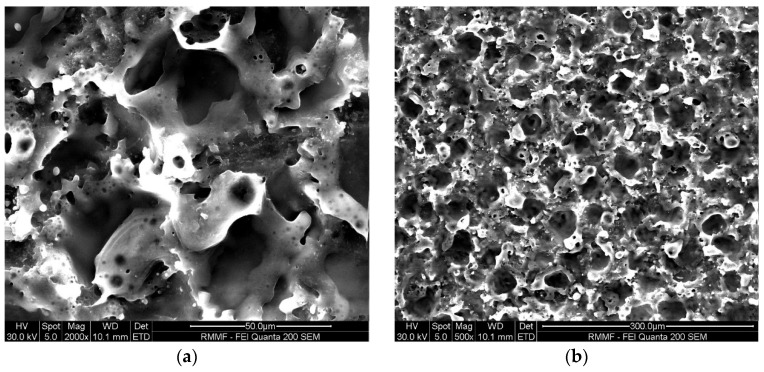
SEM images of electrode surfaces after EDM using silicon electrodes: (**a**) 2000× magnification; and (**b**) 500× magnification.

**Figure 11 micromachines-16-00147-f011:**
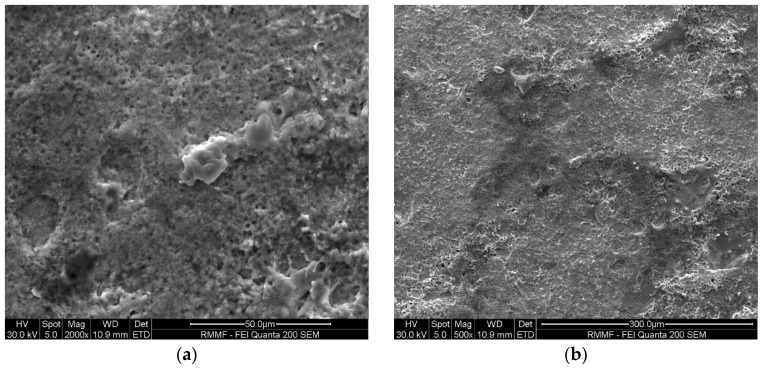
SEM images of electrode surfaces after EDM using copper electrodes: (**a**) 2000× magnification; and (**b**) 500× magnification.

**Figure 12 micromachines-16-00147-f012:**
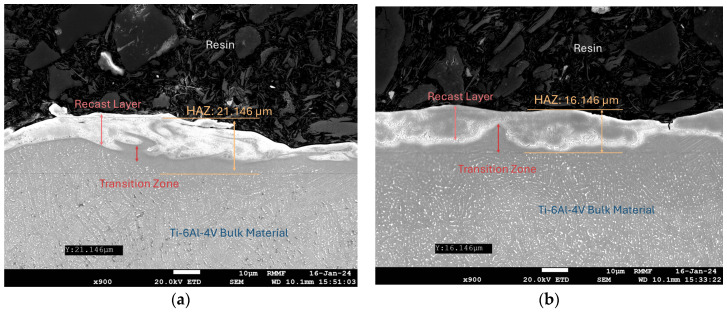
Workpiece profile after machining with different electrodes: (**a**) workpiece section after machining with copper electrodes; and (**b**) workpiece section after machining with silicon electrodes.

**Figure 13 micromachines-16-00147-f013:**
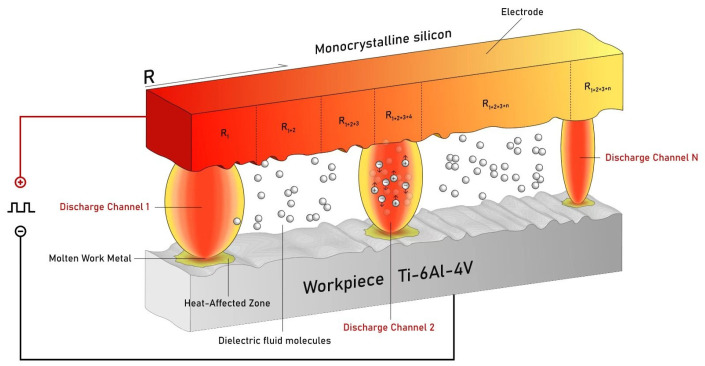
Single-electrode multi-channel discharge schematic.

**Figure 14 micromachines-16-00147-f014:**
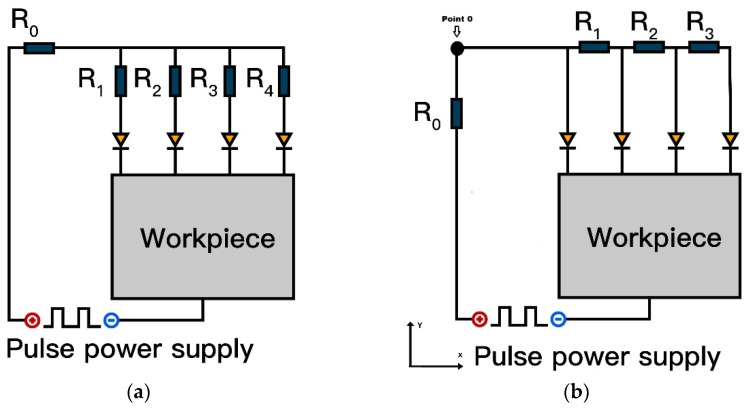
Circuit schematic for single and discrete electrodes: (**a**) discharge circuit diagram for discrete electrodes; and (**b**) discharge circuit diagram for single electrode.

**Figure 15 micromachines-16-00147-f015:**
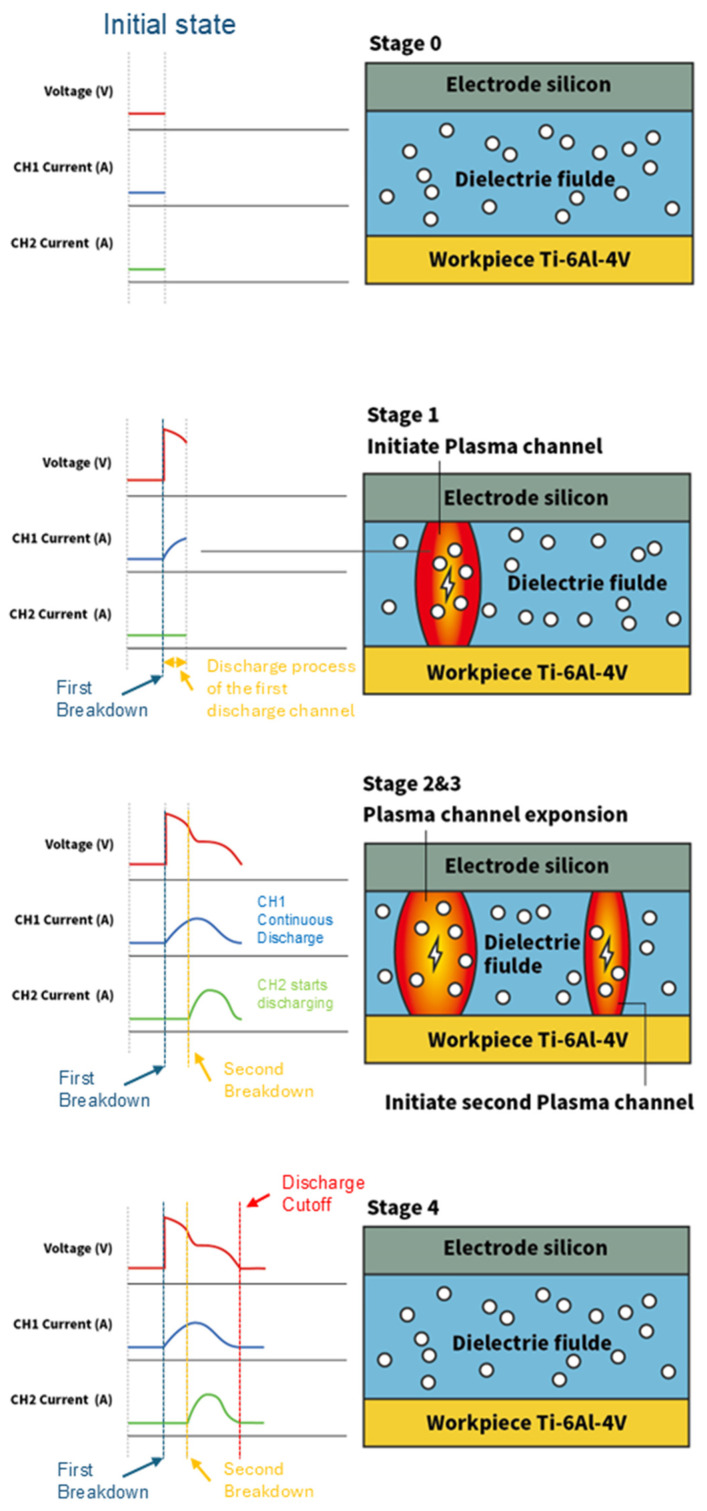
EDM processing of plasma channels for semiconductor electrodes.

**Figure 16 micromachines-16-00147-f016:**
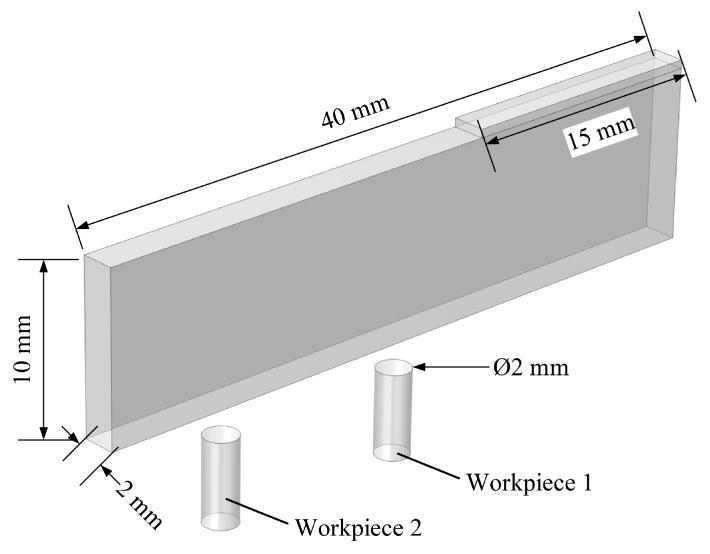
Dimensional diagram before discharge.

**Figure 17 micromachines-16-00147-f017:**
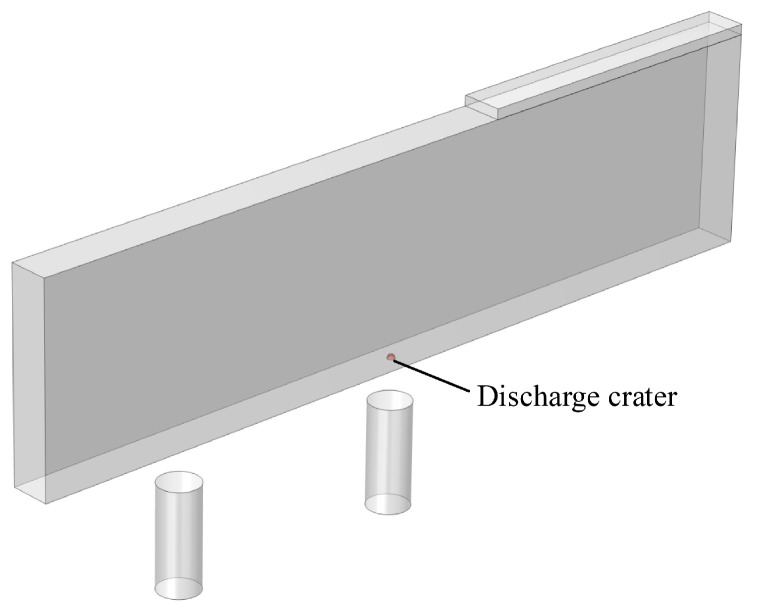
Dimensional diagram after discharge.

**Figure 18 micromachines-16-00147-f018:**
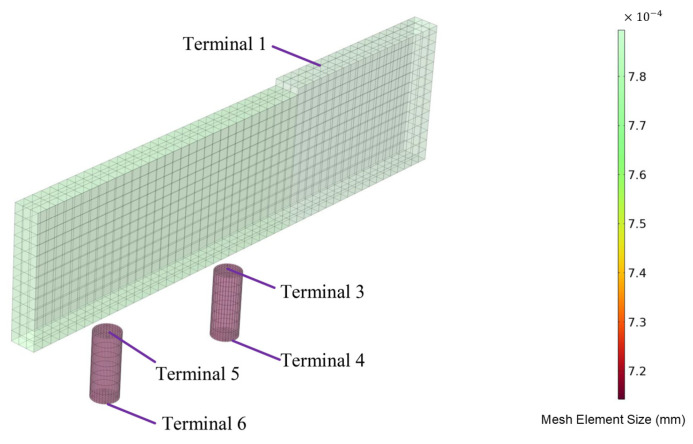
Mesh topology before discharge.

**Figure 19 micromachines-16-00147-f019:**
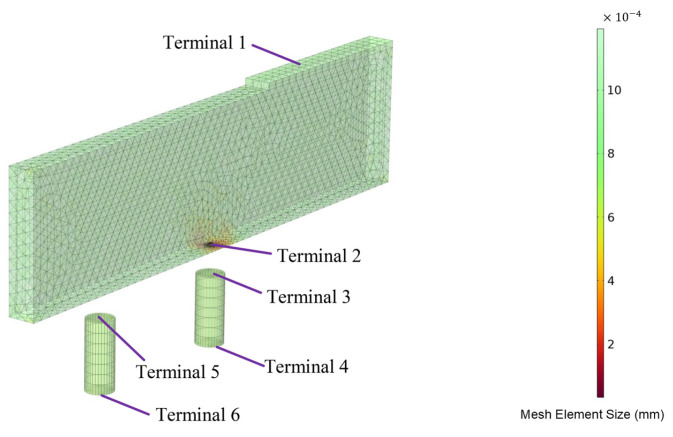
Mesh topology after discharge.

**Figure 20 micromachines-16-00147-f020:**
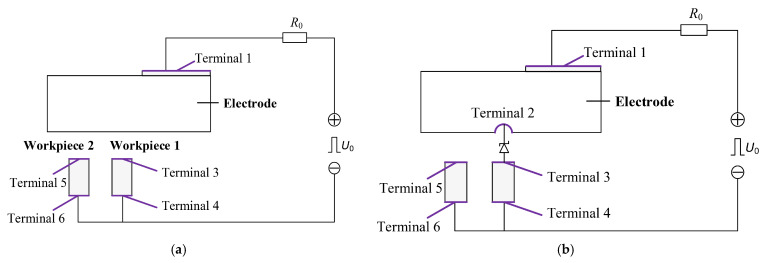
(**a**) Analog circuit diagram before discharge. (**b**) Analog circuit diagram after discharge.

**Figure 21 micromachines-16-00147-f021:**
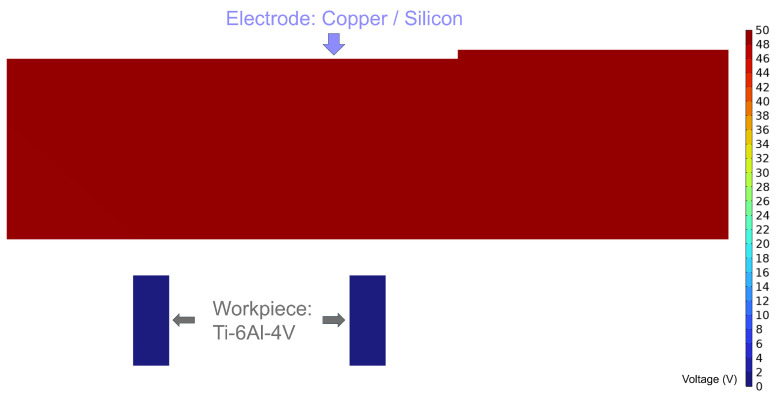
Schematic of the internal voltage of the electrode before discharge.

**Figure 22 micromachines-16-00147-f022:**
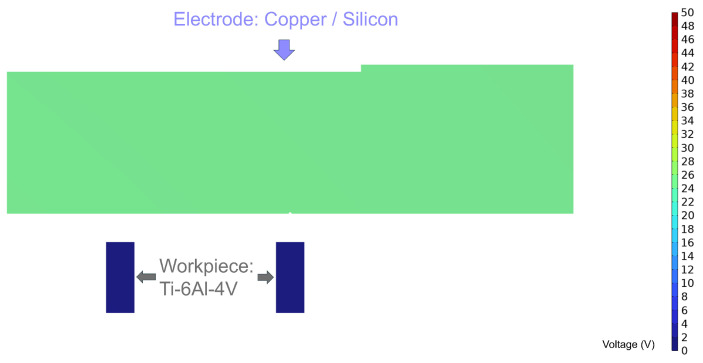
Schematic diagram of the internal voltage of a discharged copper electrode.

**Figure 23 micromachines-16-00147-f023:**
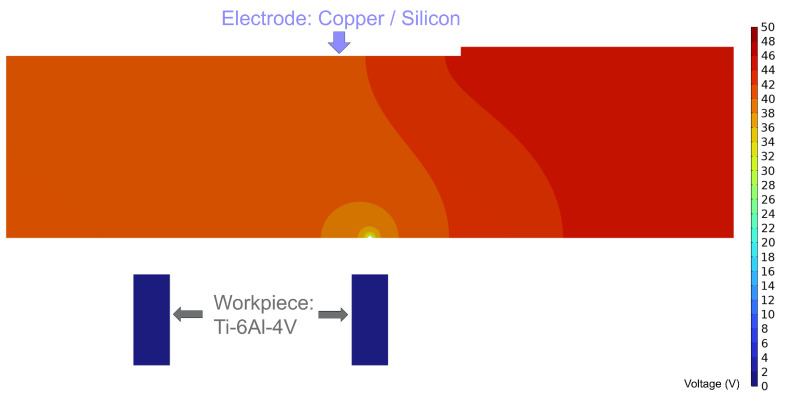
Schematic of the internal voltage of the silicon electrode after discharge.

**Table 1 micromachines-16-00147-t001:** Chemical composition of the Ti-6Al-4V workpiece.

Elements	Al	C	V	Mn	Fe	N	H	O	Ti
Composition %	6.28	0.008	4.24	0.0058	0.09	0.008	0.001	0.138	Remaining

**Table 2 micromachines-16-00147-t002:** Parameters of the verification experiment.

Item	Parameter
Workpiece material	Ti-6Al-4V bar
Machining polarity	Workpiece (-)
Electrode	Intrinsic monocrystalline Silicon/Copper
Dielectric fluid	EDM 244 (Manufacturer: Quaker Houghton, Moorabbin, VIC, Australia)
Pulse width (μs)	20
Machining voltage (V)	50
Machining current (A)	3
Response time of hall sensor (ns)	≤23
Rise time of Oscilloscope (ns)	≤3.5

**Table 3 micromachines-16-00147-t003:** Components of workpiece surface after being machined with different electrodes.

Elements	Content (wt.%)	
	Silicon Electrode	Copper Electrode
Ti	29.9	32.6
Cu		3.4
Si	7.5	
V	1.2	2.5
Al	2.3	2

**Table 4 micromachines-16-00147-t004:** Simulation parameters.

Workpiece material (cathode) (−)	Ti-6Al-4V
Electrode material (anode) (+)	Silicon, Copper
Radius of discharge crater (mm)	0.1
Silicon resistivity (Ω·cm)	1
Open-circuit voltage (V)	50
Current-limiting resistor (Ω)	5

## Data Availability

The original contributions presented in the work are included in the article: further inquiries can be directed to the corresponding author.

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
