# Peer review of "Multi-Channel Electrical Discharge Machining of Ti-6Al-4V Enabled by Semiconductor Potential Differences"

_micromachines, 2025, doi:10.3390/mi16020147_

Round 1
Reviewer 1 Report
Comments and Suggestions for Authors
The manuscript discusses the study of multi-channel EDM machining of Ti-6Al-4V with silicon electrode, which represents an interesting and novel method. This study indicates that employing a silicon semiconductor electrode could improve the surface quality, which may be helpful to the engineering applications. Here are some suggestions for the improvement of the manuscript.
1. In the introduction, it is necessary to make the expression “the high-temperature plasma” more specific and clearer.
2. The parameters set in the simulations within the manuscript are significantly different from those used in the experimental research. This is not logical, and an explanation for these differences should be provided.
3. The relationship between sections 4.1 and 4.2 of the manuscript and other sections seems much weak. Please explain the reasons, or address the contributions of the above-mentioned two sections to this manuscript.
4. The descriptions of the fourth and sixth items in the conclusion appear to lack precision, do not fully align with the preceding analysis in the manuscript, and omit certain restrictive conditions. Suggest supplementing its expression completely.
5. Suggest modifications to the size, chart type, and corresponding textual descriptions of certain graphs, particularly those that involve data comparison relationships.
Comments on the Quality of English Language
It is suggested to revise the manuscript in accordance with the standards and principles of scientific writing.
Author Response
- In the introduction, it is necessary to make the expression “the high-temperature plasma” more specific and clearer.
Response: As advised, this issue has been addressed in detail in lines 29–34, in the revised manuscript, where an explanation of its origin and specific temperature range (8000–12000 K) have been provided.
- The parameters set in the simulations within the manuscript are significantly different from those used in the experimental research. This is not logical, and an explanation for these differences should be provided.
Response: Thank you for pointing out this issue. The high-voltage scheme adopted in the simulation was to maximize the demonstration effect. However, during the actual experiments, the high voltage resulted in abnormal wear of the silicon electrodes. Consequently, the voltage was reduced for the subsequent processing experiments.
Response: In line with the reviewer’s comment, the simulation section has been revised, and Figures 18–25 and the content in Table 4 have been updated.
- The relationship between sections 4.1 and 4.2 of the manuscript and other sections seems much weak. Please explain the reasons or address the contributions of the above-mentioned two sections to this manuscript.
Response: In accordance with the reviewer’s comment, Sections 4.1 and 4.2 have been extensively revised: the redundant introduction and statements have been removed, and some relevant descriptions were moved to other sections (refer to lines 311 to 334). In the updated section, the mechanisms of achieving multi-channel discharging, specifically the instantaneous circuit state and the equivalent circuit model, were elaborated. Moreover, the comparison with traditional multi-electrode model were conducted to highlight the underlying principles and advantages.
- The descriptions of the fourth and sixth items in the conclusion appear to lack precision, do not fully align with the preceding analysis in the manuscript, and omit certain restrictive conditions. Suggest supplementing its expression completely.
Response: Thank you for your suggestion. The fourth and sixth conclusions have been further refined and updated in lines 447–450 and 456–460 of the revised manuscript.
- Suggest modifications to the size, chart type, and corresponding textual descriptions of certain graphs, particularly those that involve data comparison relationships.
Response: As advised by the reviewer, the entire paper has been thoroughly checked, and Figures 18, 19, 21, 22, and 23 have been revised by adding annotations for the physical quantities and their corresponding units, making them easier to understand.
Reviewer 2 Report
Comments and Suggestions for Authors
Dear Authors,
the paper is interesting, but requires some improvements before can be accepted. Please follow the comments listed below.
1. Lines 29-32 - The authors reported that EDM causes surface defects. Yes, craters are formed, which, as the authors wrote, affect wear resistance or fatigue strength. However, it is worth adding in this paragraph of the paper that a rough surface after EDM is sometimes desirable in order to improve the hydrophobic properties of the surface, or generally speaking, to change the wettability and lubricity of the surface. Please refer to https://doi.org/10.3390/ma17235716
2. Line 36 - please give examples of thermal defects and what effects they may have.
3. Lines 42-43 - discharge energy depends on a larger number of parameters. Not only current and time, but also voltage.
4. Table 2 - please add manufacturer of dielectric liquid.
5. Fig. 3 - please specify what worpiece 1 and worpiece 2 are (in the figure), so that the reader does not have to look for it in the text.
6. Fig. 3 - there is a very large MRR of silicone electrodes. Have the authors tried to make a cost calculation? A precise indication of the economic benefits of using silicone electrodes would be advisable.
7. Chapter 3.2 - Typically, these roughness values ​​are given in micrometers for Ra. This improves readability.
8. Fig. 9 - After using silicone electrodes, the surface chemistry is different than in the case of copper. This is obvious. However, the question is about applications. Will changing the surface chemistry not limit certain applications? Please indicate the applications of titanium alloy in terms of surface chemistry. For example, how does this relate to biomedical applications?
9. Fig 20, 21, 23, 24, 25 - What does the scale represent, and give the units.
Author Response
- Lines 29-32 - The authors reported that EDM causes surface defects. Yes, craters are formed, which, as the authors wrote, affect wear resistance or fatigue strength. However, it is worth adding in this paragraph of the paper that a rough surface after EDM is sometimes desirable in order to improve the hydrophobic properties of the surface, or generally speaking, to change the wettability and lubricity of the surface. Please refer to https://doi.org/10.3390/ma17235716
Response: The authors appreciate the reviewer’s comment. As advised, more discussions about the surface quality caused be craters have been added (lines 37 – 39) and the new reference recommended by the reviewer has been cited in the context [8].
- Line 36 - please give examples of thermal defects and what effects they may have.
Response: As advised, three more examples have been added in lines 42–46, along with explanations of their impact on the workpiece.
- Lines 42-43 - discharge energy depends on a larger number of parameters. Not only current and time, but also voltage.
Response: We agree with the reviewer. The voltage and polarity have been added to line 53 of the revised manuscript.
- Table 2 - please add manufacturer of dielectric liquid.
Response: As advised, the manufacturer of the dielectric liquid, ‘Quaker Houghton,’ has been added to the corresponding position in Table 2.
- 3 - please specify what worpiece 1 and worpiece 2 are (in the figure), so that the reader does not have to look for it in the text.
Response: As advised, the workpiece "Ti-6Al-4V" has been labeled for Workpieces 1 and 2 in Figure 3.
- 3 - there is a very large MRR of silicone electrodes. Have the authors tried to make a cost calculation? A precise indication of the economic benefits of using silicone electrodes would be advisable.
Response: The authors appreciate this comment. As the primary focus of this study is to investigate the fundamental mechanism of achieving multi-channel discharge through the bulk resistivity of electrodes rather than its application in industry productions, considerations regarding the cost of electrode materials have been relatively limited. However, since this is an important issue, in line with this comment, new discussions on these issues have been added in the Conclusion and Further Work section.
- Chapter 3.2 - Typically, these roughness values ​​are given in micrometers for Ra. This improves readability.
Response: The unit for Ra has been corrected to micrometers in lines 208, 209, 215, 219, and 220.
- 9 - After using silicone electrodes, the surface chemistry is different than in the case of copper. This is obvious. However, the question is about applications. Will changing the surface chemistry not limit certain applications? Please indicate the applications of titanium alloy in terms of surface chemistry. For example, how does this relate to biomedical applications?
Response: The reviewer is correct, and this is indeed a valuable topic for discussion. The surface modification of workpieces after EDM is influenced by factors such as electrode material migration and dielectric fluid decomposition. Some studies have even intentionally utilized specialized electrode materials or added specific substances to the dielectric fluid to take advantage of this phenomenon. In general, when silicon is deposited on the surface of titanium alloys, the effects tend to be beneficial, primarily enhancing the material's hardness and wear resistance.
While this study focuses on the underlying mechanism of multi-channel discharge, we recognize the significance of this phenomenon and have included it in the Conclusion and Further Work section.
In line with the reviewer’s comment, a new paragraph was also added in Lines 245-256 to discuss this issue.
- Fig 20, 21, 23, 24, 25 - What does the scale represent, and give the units.
Response: The physical quantities and their respective units have been added in Figures 20, 21, 23, 24, and 25.